# Multimodal Characterization of a PSMA-Positive Thyroid Nodule Using ^68^Ga-PSMA and ^124^Iodine PET/US Fusion Imaging

**DOI:** 10.3390/diagnostics12020472

**Published:** 2022-02-12

**Authors:** Martin Freesmeyer, Falk Gühne, Robert Drescher, Thomas Winkens, Nikolaus Gassler, Philipp Seifert

**Affiliations:** 1Clinic of Nuclear Medicine, Jena University Hospital, 07749 Jena, Germany; falk.guehne@med.uni-jena.de (F.G.); robert.drescher@med.uni-jena.de (R.D.); thomas.winkens@med.uni-jena.de (T.W.); philipp.seifert@med.uni-jena.de (P.S.); 2Section of Pathology, Institute of Forensic Medicine, Jena University Hospital, 07749 Jena, Germany; nikolaus.gassler@med.uni-jena.de

**Keywords:** PSMA, prostate cancer, thyroid nodule, PET/US fusion imaging, immunohistochemistry

## Abstract

A 54-year-old male diagnosed with prostate cancer was referred for ^68^Gallium-PSMA-11 PET/CT. The scan revealed a solitary PSMA-positive thyroid lesion. On PET/ultrasound fusion imaging, a nodule with moderate risk of malignancy (TIRADS 4B) could be unambiguously correlated. Additional ^124^Iodine PET/ultrasound fusion imaging revealed normal iodine uptake within the PSMA-positive thyroid nodule. Fine-needle aspiration cytology was performed using an ultrasound needle-guidance system. The cytopathological investigation confirmed a benign thyroid nodule and excluded a thyroid carcinoma as well as a prostate cancer metastasis. Immunohistochemistry was positive for thyroglobulin staining.

**Figure 1 diagnostics-12-00472-f001:**
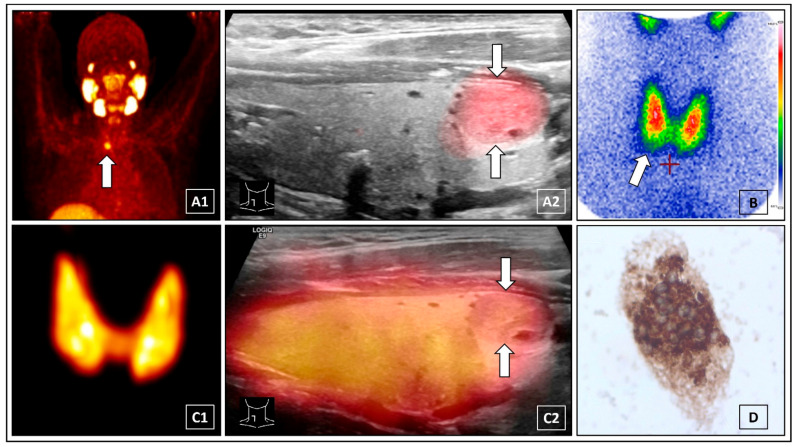
Prostate-specific membrane antigen (PSMA) “theranostics” is gaining increasing importance in the treatment of prostate cancer [1]. A 54-year-old male diagnosed with prostate cancer presented to our clinic for ^68^Gallium-PSMA-11 PET/CT. The PET-scan incidentally revealed a solitary PSMA-positive lesion of the thyroid gland (**A1**, white arrow). There was no history of thyroid pathologies and laboratory thyroidal values were normal. Subsequently, ^68^Gallium-PSMA-11 PET/ultrasound (US) fusion imaging was performed and unambiguously depicted a solitary PSMA-positive thyroid nodule (15 × 13 × 8 mm) in the lower right lobe (**A2**, white arrows), classified as Kwak-TIRADS 4B [2]. PSMA-positive thyroid uptake can be related to several diagnoses, including thyroid cancer, metastases of prostate cancer or renal cell carcinoma, benign thyroid nodules, and de Quervain’s thyroiditis [3,4,5]. ^99m^Tc-scintigraphy did not clearly characterize the nodule (**B**, white arrow) and therefore cervical ^124^Iodine PET/CT was performed. The ^124^Iodine maximum intensitiy projection (MIP) PET imagiges did not reveal any hyper- or hypofunctional thyroidal lesions (**C1**). Additional PET/US fusion imaging clearified a normal iodine uptake of the PSMA-positive nodule (**C2**, white arrows). For real-time PET/US fusion imaging, the PET/CT images (biograph mCT 40; Siemens, Erlangen, Germany) were transferred to the LOGIQ E9 ultrasound device (GE Healthcare, Milwaukee, WI, USA). According to anatomical landmarks on CT, e.g., spine, larynx, trachea, the PET/CT images were manually superimposed and aligned to the ultrasound images using a magnetic field based navigation system and the VNAV software (GE Healthcare) [6]. For ^124^Iodine PET/US fusion imaging, a single bed position (10 min scan time) low-activity (1 MBq ^124^Iodine) cervical PET/CT (low-dose CT scan) was performed. The effective whole body dose for the patient was ~6.8 mSv [7]. The “FUSION iENA” study (obtained by independend ratings of 106 nuclear medicine physicians) revealed that ^124^Iodine PET/US fusion imaging significantly improves the accuracy and the confidence of the functional assessement of thyroid nodules as well as influences the suggested treatment for patients with ambiguous findings on conventional diagnostics [8]. Fine-needle cytology was performed using an magnet-based ultrasound needle-guidance system to ensure that the cells were acquired from the PSMA-positive thyroid nodule [9]. The cytopathological investigation showed a benign thyroid nodule according to Bethesda category II with positive thyroglobulin staining (**D**) [10]. Finally, thyroid cancer as well as an intra-thyroid prostate cancer metastasis could be ruled out. This interesting image demonstrates the first application of PET/US fusion imaging in a PSMA-positive thyroid nodule and demonstrates the diagnostic potential of combined modern multimodal methods in the field of nuclear medicine.

## Data Availability

Not applicable.

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
