# Peer review of "Multimodal Characterization of a PSMA-Positive Thyroid Nodule Using 68Ga-PSMA and 124Iodine PET/US Fusion Imaging"

_diagnostics, 2022, doi:10.3390/diagnostics12020472_

Round 1
Reviewer 1 Report
This Interesting Image can be the new and interesting one. The format of the article can be improved.
Reviewer 2 Report
The paper under evaluation is an interesting image, describing the case of a 54-year-old male, referred to 68Ga-PSMA-11 PET/CT for prostate cancer, presenting an unusual focal uptake in a thyroid nodule. The patient was submitted to 99mTc-pertechnetate scintigraphy and 124Iodine PET/CT. Final diagnosis, after having performed PET/US imaging fusion and PET-guided FNAB, the suspected finding resulted in a benign thyroid nodule.
Some concerns:
- the authors state that they have performed 124Iodine PET but they did not show any 124I-PET image, except for that related to the fusion PET/US. Please add 124I-PET whole body image.
- My major concern regards the modality of fusion between PET and US, that should be briefly described. While US is a 2D imaging modality, PET is a 3D imaging method. How did they perform this fusion? Which software was used for co-registration? Did the authors use some anatomical references? A co-registration simply deriving from image superimposition seems to me too operator-dependent.
- PET-PSMA false positivity in benign nodules has been described (doi: 10.1186/s40644-018-0175-3).
Reviewer 3 Report
Case report to exclude thyroid cancer (prostate metastasis or thyroid follicular or papillomatous cancer) in a patient with suspected PSMA positive nodule.
Short description, adequate for such a case report.
A few more words explaining "124Iodine-PET/US fusion imaging was performed since this investigation improves the functionality assessment of ambiguous thyroid nodules (5)." : could you give some data on this technic? Is it already available in many nuclear medicine departements in Europa? A few words of summary of reference 5 : sensitiviy / specificity / predictive negative value / predictive positive value of this technic? How many indications (which percentage of "thyroid nodules" could benefit of this technique?
Author Response
Dear valued Reviewer,
We thank you very much for taking the time to carefully read our manuscript and helping us to further improve it through the valuable comments you have provided. Your feedback helped us considerably to resubmit a better and more understandable manuscript.
Please find below our detailed response to each of your comments.
Round 2
Reviewer 2 Report
The authors have properly addressed Reviewer's suggestions.
In particular, I have appreciated how the authors thoroughly explained PET/US coregistration procedure.